# Soluble collectin-12 mediates C3-independent docking of properdin that activates the alternative pathway of complement

Jie Zhang[1,2], Lihong Song[1,3], Dennis V Pedersen[4], Anna Li[1,2], John D Lambris[5], Gregers Rom Andersen[4], Tom Eirik Mollnes[6,7,8], Ying Jie Ma[1]*, Peter Garred[1]*

[1]The Laboratory of Molecular Medicine, Department of Clinical Immunology, Faculty of Health and Medical Sciences, University of Copenhagen, Copenhagen, Denmark; [2]Department of Clinical Pharmacy, School of Basic Medicine and Clinical Pharmacy, China Pharmaceutical University, Nanjing, China; [3]Department of Pharmaceutical Science, School of Pharmacy, Shenyang Pharmaceutical University, Shenyang, China; [4]Department of Molecular Biology and Genetics, Center for Structural Biology, Aarhus University, Aarhus, Denmark; [5]Department of Pathology and Laboratory Medicine, Perelman School of Medicine, University of Pennsylvania, Philadelphia, United States; [6]Department of Immunology, Oslo University Hospital, and University of Oslo, Oslo, Norway; [7]Research Laboratory, Nordland Hospital, K. G. Jebsen TREC, University of Tromsø, Bodø, Norway; [8]Center of Molecular Inflammation Research, Norwegian University of Science and Technology, Trondheim, Norway

*For correspondence:
mayingjie606@hotmail.com (YJM);
peter.garred@regionh.dk (PG)

**Abstract** Properdin stabilizes the alternative C3 convertase (C3bBb), whereas its role as pattern-recognition molecule mediating complement activation is disputed for decades. Previously, we have found that soluble collectin-12 (sCL-12) synergizes complement alternative pathway (AP) activation. However, whether this observation is C3 dependent is unknown. By application of the C3-inhibitor Cp40, we found that properdin in normal human serum bound to *Aspergillus fumigatus* solely in a C3b-dependent manner. Cp40 also prevented properdin binding when properdin-depleted serum reconstituted with purified properdin was applied, in analogy with the findings achieved by C3-depleted serum. However, when opsonized with sCL-12, properdin bound in a C3-independent manner exclusively via its tetrameric structure and directed in situ C3bBb assembly. In conclusion, a prerequisite for properdin binding and in situ C3bBb assembly was the initial docking of sCL-12. This implies a new important function of properdin in host defense bridging pattern recognition and specific AP activation.

## Introduction

Properdin was originally described in 1954 by Pillemer et al. as a serum component that promotes complement activation in an antibody-independent manner (*Pillemer et al., 1954*). Since then, controversies on how properdin facilitates complement activation have encircled. The first of the two common conceptions are well accepted. Properdin binds to and stabilizes the alternative C3 convertase C3bBb, thereby extending its half-life 5- to 10-fold and inhibit its degradation by factor I (*Fearon and Austen, 1975*; *Schreiber et al., 1975*; *Medicus et al., 1976*). However, since challenged in 1958 by Nelson et al., who suggested that the binding of properdin requires the presence

of activator-bound C3b, the second conception on the role of properdin as a pattern-recognition molecule (PRM) leading to the alternative pathway (AP) activation has been a matter of dispute (*Nelson, 1958*; *Kemper et al., 2010*).

Recently, the concept of properdin as a PRM emerged from a series of studies designed to understand whether properdin could initiate and direct in situ C3 convertase assembly. Interest in this concept was renewed when recombinant transmembrane properdin in HEK-293 cells was shown to initiate AP activation (*Vuagnat et al., 2000*) and surface plasmon resonance methodology confirmed that biosensor chip bound properdin can provide a platform to initiate C3bBbP assembly from purified C3b, factor B and factor D (*Hourcade, 2006*). In line with those findings, several recent observations have also supported the recognition role of properdin in several models of molecular patterns including pathogens, endogenous cells and various biological substrates (*Cortes et al., 2011*; *Kemper et al., 2008*; *Xu et al., 2008*; *Saggu et al., 2013*; *O'Flynn et al., 2014*; *Wang et al., 2015*). However, Harboe et al. recently challenged the concept of properdin as a PRM and argued that properdin binding to complement activating surfaces expressing pathogen- and damage-associated molecular patterns, including zymozan, *Escherichia coli*, *Neisseria meningitidis,* myeloperoxidase and umbilical vein cells, is strictly dependent on initial C3 activation with subsequent binding of properdin to stabilize C3bBb (*Harboe et al., 2012*; *Harboe et al., 2017*).

Collectin-12 (CL-12; also known as collectin placenta one or CL-P1) is a newly identified collagen-like molecule with multimeric structure assembled by ~140 kDa subunits and mainly present as a monomer, dimer or trimer (*Nakamura et al., 2001*; *Ohtani et al., 2001*; *Ma et al., 2015*). CL-12 is characterized by an intracytoplasmic domain, a transmembrane domain and an ectodomain subdivided into a coiled-coil domain, a collagen-like domain and a carbohydrate-recognition domain. The ectodomain is crucial for oligomeric assembly and recognition of various pathogenic or endogenous molecular patterns (*Nakamura et al., 2001*; *Ohtani et al., 2001*; *Mori et al., 2014*). CL-12 was originally defined as scavenger receptor C-type lectin because it shares structural and functional similarities with type A scavenger receptor (*Nakamura et al., 2001*; *Ohtani et al., 2001*). CL-12 is mainly expressed as transmembrane molecules on cells originating from the endothelium and placenta but is also found on gastric stromal cells and phagocytes such as alveolar macrophages, central nervous system resident macrophages and microglia cells. CL-12 has been shown to mediate scavenging properties in sequestration of damage-associated molecular patterns such as oxidized low-density lipoprotein (*Nakamura et al., 2001*; *Ohtani et al., 2001*; *Selman et al., 2008*). Furthermore, CL-12 binds glycans bearing both terminal galactose and fucose moieties and presents a striking high affinity for Lewis[X] trisaccharides (*Coombs et al., 2005*).

Lewis[X] trisaccharides are commonly displayed on adhesion molecules of various leukocytes, suggesting that CL-12 could modulate leukocyte recruitment (*Coombs et al., 2005*; *Elola et al., 2007*). CL-12 has also been shown to bind fibrillar β-amyloid protein, thus indicating a potential role in scavenging of amyloid (*Nakamura et al., 2006*). More recent data show that CL-12 is involved in myelin internalization by central nervous system resident phagocytes (*Bogie et al., 2017*). Moreover, CL-12 as a C-type lectin exhibits innate immune defense characteristics mediating phagocytosis of certain bacteria and fungi (*Ohtani et al., 2001*; *Jang et al., 2009*; *Zhang et al., 2019*). Further, in a recent study Chang et al. suggested the importance of CL-12 as a C-type lectin receptor involved in *H. pylori*-gastric stromal cells interaction and mediating crosstalk between these cells and dendritic cells (*Chang et al., 2018*). Finally, and particularly important for the background of this study, surface-bound CL-12 has been suggested to play a role in complement activation through the classical pathway via its association with C-reactive protein and C1q (*Roy et al., 2017*).

In previous work, we showed the existence of a soluble form of CL-12 in addition to the transmembrane form and revealed its specific opsonic properties towards clinically important fungal pathogens (*Ma et al., 2015*; *Zhang et al., 2019*). Interestingly, we found that soluble CL-12 could synergize AP activation upon opsonization of *A. fumigatus* leading to activation of C3 and the terminal membrane attack complex formation (*Ma et al., 2015*). Thus, the present study aimed to investigate whether and how soluble CL-12 opsonizing *A. fumigatus* interacts with properdin in a C3-independent manner and leads to specific AP activation.

## Results

### C3b and properdin deposition on *A. fumigatus* incubated in normal human serum was C3-dependent

Compstatin is a potent peptide inhibitor of C3 activation, which has been further modulated to a more potent analog named Cp40 (*Risitano et al., 2014*). In order to study our hypothesis, *Aspergillus fumigatus* (*A. fumigatus*) was used as a model of infection. Normal human serum (NHS) was used as a complement source and C3b deposition and properdin binding were determined on *A. fumigatus* with or without the addition of Cp40. Cp40 completely inhibited C3b deposition at 1.5 µM final concentration (*Figure 1A*). Concomitant with this observation, properdin binding was completely abolished in the presence of the Cp40 (6 µM) (*Figure 1B*). A control peptide had no effects on C3b or properdin binding (*Figure 1A–B*).

### Soluble CL-12-induced properdin binding was not dependent on initial C3b deposition

First, *A. fumigatus* was incubated with purified CL-12, and binding of CL-12 was measured by flow cytometry (*Figure 2A*). Second, *A. fumigatus* was incubated with NHS, and the deposition of properdin was found to be markedly increased on the fungi opsonized with CL-12 than the non-opsonized fungi (*Figure 2B*). Third, to further validate our previous findings that soluble CL-12 triggers AP

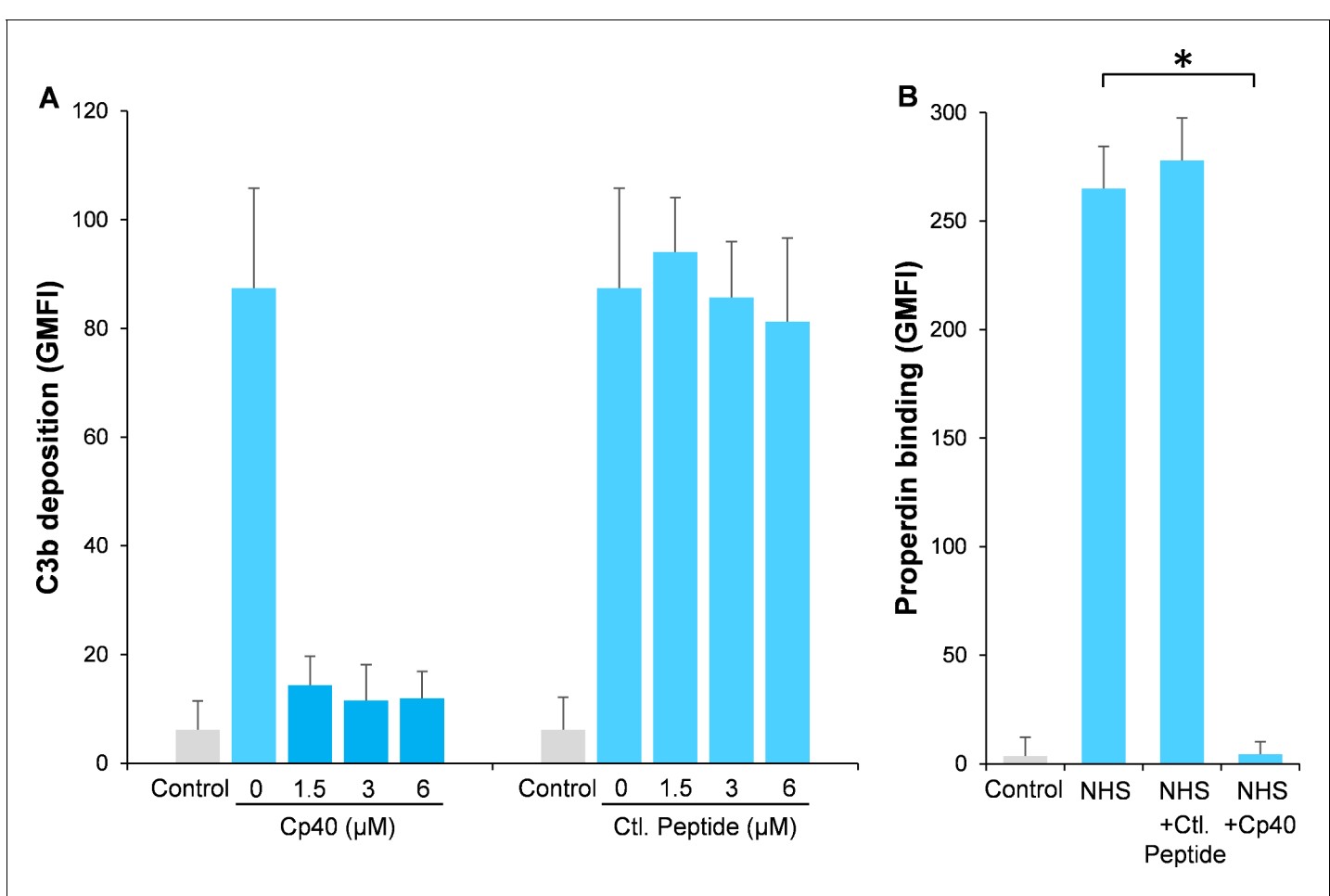

**Figure 1.** Inhibition of complement activation by compstatin analog Cp40. NHS (20%) was preincubated with compstatin analog Cp40 or its control peptide (A: 0 ~ 6 µM; B: 6 µM) prior to inducing complement activation on *A. fumigatus* through incubation in the presence of $Ca^{2+}$. C3b deposition (A) or properdin binding (B) was analyzed and expressed as the geometric mean fluorescence intensity (GMFI) by flow cytometry. Data are expressed as mean ± S.E.M from three independent experiments. Results are representative of at least six independent experiments. *p<0.01.

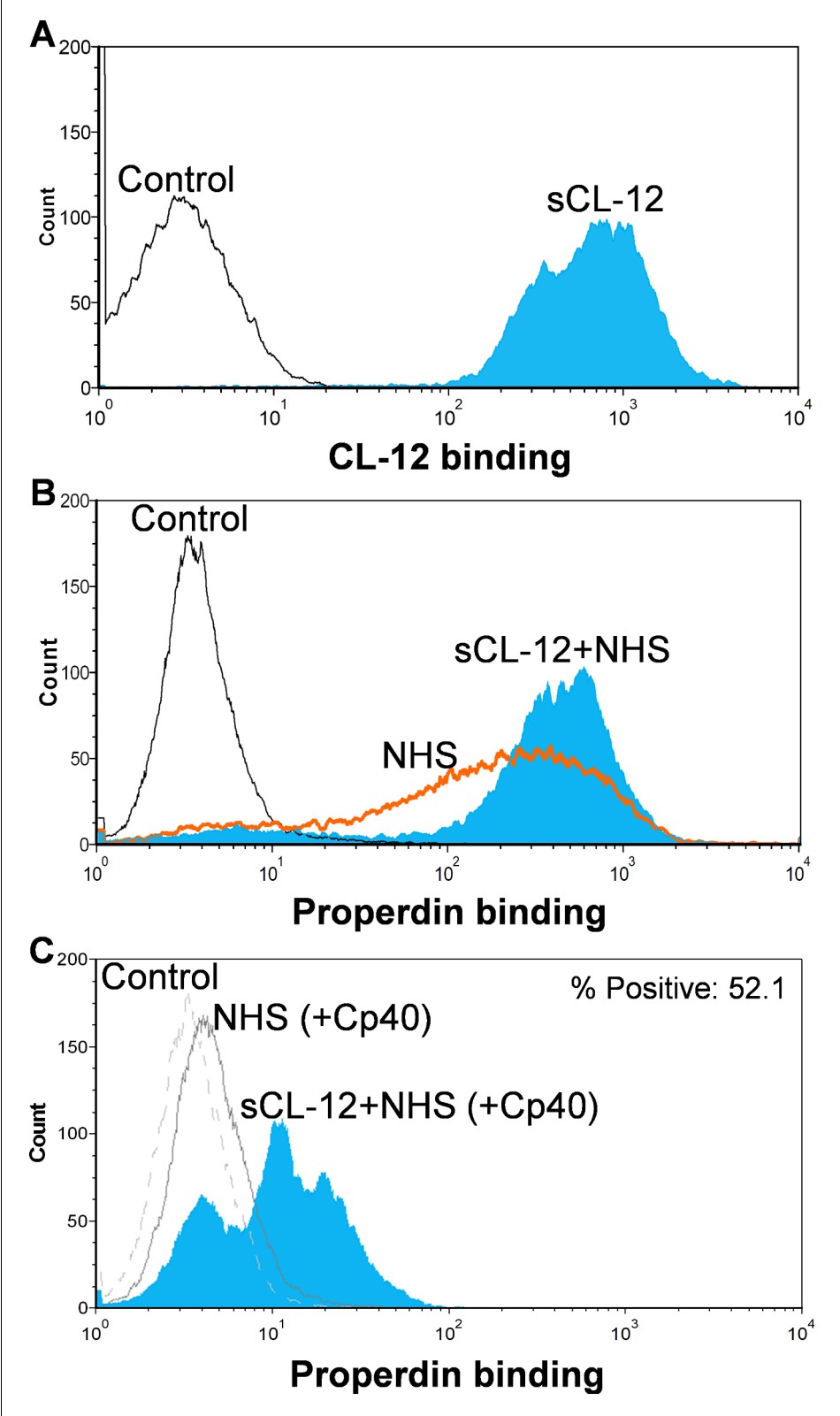

**Figure 2.** Binding of properdin to *A.fumigatus* in the presence of soluble CL-12 and compstatin analog Cp40. *A. fumigatus* were incubated with or without soluble CL-12 (sCL-12) (5 μg/ml) prior to addition of NHS (20%) under EGTA-Mg$^{2+}$ buffer. In some experiments, NHS (20%) was preincubated with compstatin analog Cp40 (6 μM) prior to induce complement activation. CL-12 (**A**) or properdin (**B, C**) binding was analyzed by flow cytometry. Results are representative of at least six independent experiments.

complement activation via properdin, C3 activation was inhibited by Cp40 and the effect of soluble CL-12 on properdin recruitment was evaluated. In the presence of Cp40, properdin binding from NHS alone was completely abolished, whereas properdin from NHS was still able to bind CL-12-pre-opsonized fungi, although to a lesser extent than without Cp40. (*Figure 2C*).

## Soluble CL-12 mediated C3b-independent binding of properdin

To further investigate the CL-12-dependent properdin binding, properdin-depleted serum (fP-Dpl) was reconstituted with serum properdin in the absence and presence of Cp40. In the absence of Cp40, no binding of properdin was observed in fP-Dpl, whereas after reconstitution the binding of properdin was comparable to NHS as described above (*Figure 3A*). Pre-opsonization with CL-12 markedly enhanced the properdin binding in reconstituted serum (*Figure 3A*). C3 inhibition by Cp40 again virtually abolished the properdin binding to the fungi in the absence of CL-12, whereas it was restored solely upon pre-opsonization with soluble CL-12 (*Figure 3B*).

## The binding of properdin to CL-12-opsonized fungi was specifically C3-independent

To further substantiate the exclusive effect of soluble CL-12 on C3b-independent binding of properdin, C3- depleted serum (C3-Dpl) was used as complement source to exclude the influence of C3b deposition and properdin binding was determined with a series of serial dilution of the serum (~10–100%). Reconstitution with exogenous C3 under physiological concentration was able to restore AP activation as validated by determination of C3b deposition using the EGTA-Mg$^{2+}$ containing buffer (*Figure 4A*). Again, when *A. fumigatus* was opsonized with soluble CL-12, properdin binding was markedly increased in a serum concentration-dependent manner, (*Figure 4B*), thus reflecting the increase in properdin concentration with constant depleted C3. This was also confirmed through application of 100% NHS (*Figure 4C–E*), where C3 activation (*Figure 4C*) and properdin binding (*Figure 4E*) were completely removed by addition of the Cp40.

To further confirm the prerequisite of soluble CL-12 for the C3b-independent binding of properdin leading to complement activation observed with *A. fumigatus,* we turned to *Aspergillus niger* (*A. niger*) as an another model of infection based on our recent findings (*Zhang et al., 2019*). In agreement with the results for *A. fumigatus*, soluble CL-12-mediated properdin binding was detected in NHS in the presence of Cp40 (*Figure 4—figure supplement 1*) or fP-Dpl plus exogeneous properdin (*Figure 4—figure supplement 2*) and was repeated with the use of C3-Dpl serum (*Figure 4—figure supplement 3*). By contrast, when *Candida albicans* (*C. albicans*) was applied, no apparent binding of soluble CL-12 was noted (*Zhang et al., 2019*) and complement was only activated in mannose-binding lectin (MBL) (or collectin-11)-dependent manner (*Ma et al., 2013a*).

## Soluble CL-12 solely recruits properdin with high oligomeric structure

To identify which structural organization is required for the interaction of soluble CL-12 with the oligomeric forms of properdin, recombinant wild type properdin was produced in HEK293F cells and separated by size-exclusion chromatography (SEC) in two fractions: (i) containing dimers and trimers (fP DT) and (ii) containing tetramers (fP T) (*Pedersen et al., 2017*). Furthermore, a monomer-like properdin (fP M) was manipulated and obtained through co-expression of separate N-terminal (TB-TSR3) and C-terminal (TSR4-TSR6) constructs (*Figure 5A*; *Pedersen et al., 2019a*). The purity and oligomerization status of these properdin preparations were confirmed by SDS-PAGE (*Figure 5B*, *inset*) and by SEC analysis (*Figure 5B*). In the SDS-PAGE analysis, both fP T and fP DT migrated with gel mobility corresponding to ~53 kDa in parallel with purified serum properdin, whereas the monomer-like properdin comprising the two fragments TB-TSR3 and TSR4-TSR6 appeared as two bands corresponding to ~33 kDa and ~31 kDa, respectively (*Figure 5B*, *inset*). Compared with purified serum properdin organized mainly as dimer, trimer and tetramer, SEC analysis revealed that the used fP DT contained approx. 50% dimers and 50% trimers eluting in two separate peaks, the fP T was primarily tetrameric with a low content of trimer, while the fP M eluted in a single peak in line with our previous observations (*Figure 5B*; *Pedersen et al., 2017*; *Pedersen et al., 2019a*). All three properdin variants recognized C3b deposited on the fungus and induced C3 amplification via stabilization of the C3 convertase (*Figure 5C*), consistent with our previous findings (*Pedersen et al., 2017*; *Pedersen et al., 2019a*). Consistently, pre-opsonization with CL-12 enhanced the binding of

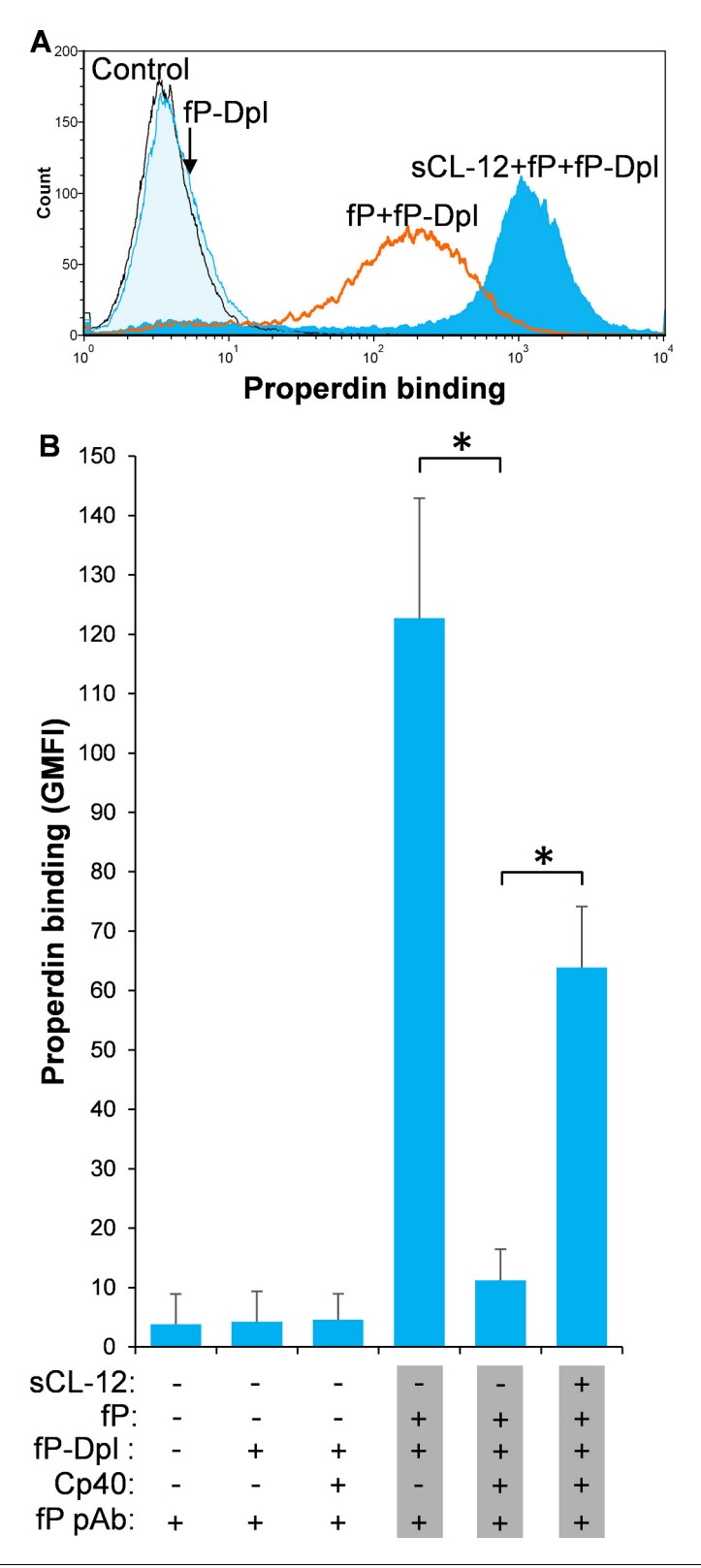

**Figure 3.** Soluble CL-12-dependent properdin binding on *A.fumigatus*. *A. fumigatus* were incubated with or without sCL-12 (5 µg/ml) prior to addition of properdin-depleted serum (fP-Dpl) (10%) reconstituted with serum properdin (10 µg/ml) (**A**). In some experiments, the serum was preincubated with or without compstatin analog Cp40 (6 µM) under EGTA-Mg$^{2+}$ buffer (**B**). Properdin binding was analyzed by flow cytometry. The GMFI was used to

*Figure 3 continued on next page*

*Figure 3 continued*

assess protein binding. B is expressed as mean ± S.E.M from three independent experiments. Results are representative of at least six independent experiments. *p<0.01.

fP T dose-dependently in the presence of Cp40 (*Figure 5D*). In contrast, no clear binding was detected when adding the fP DT or the fP M instead (*Figure 5D*). These results obtained with *A. fumigatus* were confirmed by parallel experiments with *A. niger* (*Figure 5E* and *Figure 5—figure supplement 1*).

## Soluble CL-12-induced properdin binding directs in situ assembly of the AP C3 convertase

To further validate whether surface-bound properdin directs C3bBb assembly and whether prior opsonization of CL-12 indeed amplifies in situ assembly of C3bBb via properdin recruitment, Bb deposition was assessed on *A. fumigatus*. In contrast to properdin-deficient serum, reconstitution of exogenous properdin sustained C3b-dependent Bb association, since no Bb deposition was observed when Cp40 was present (*Figure 6*). In accordance with the previous properdin binding data, pre-opsonization with CL-12 dramatically increased the Bb deposition in the reconstituted serum, while C3 inhibition by Cp40 again abolished the Bb deposition (*Figure 6*). Consistent results were also obtained on *A. niger* (*Figure 6—figure supplement 1*).

## Discussion

Properdin is well established as a stabilizer of the AP convertase (*Fearon and Austen, 1975*; *Schreiber et al., 1975*; *Medicus et al., 1976*). We have previously shown that properdin does not recognize pathogenic or endogenous molecular patterns unless C3 is activated (*Harboe et al., 2012*; *Harboe et al., 2017*). Here we, to the best of our knowledge, for the first time show that properdin might bind secondarily to a PRM, CL-12, opsonized on *A. fumigatus*, in a C3-independent manner as illustrated in *Figure 7*.

CL-12 is a newly identified scavenger receptor C-type lectin and mainly expressed not only on endothelial and placental cells, but also on gastric stromal cells and some phagocytes (*Nakamura et al., 2001*; *Ohtani et al., 2001*; *Selman et al., 2008*; *Chang et al., 2018*). Recent data have shown that some of the macrophage-specific scavenger receptors, for instance, scavenger receptor/collagen-like domain 163 (CD163) and endocytic mannose receptor (MR, CD206), can be found as soluble proteins produced by proteolytic cleavage of the receptors from the cell membrane during inflammation and by macrophage activation (*Etzerodt and Moestrup, 2013*; *Møller, 2012*). Both CD163 and CD206 have recently been detected in blood samples, and the levels have been found to be closely associated with several disease states including infections and some liver diseases (*Møller, 2012*; *Rødgaard-Hansen et al., 2014*; *Andersen et al., 2018*; *Loonen et al., 2019*; *Rødgaard-Hansen et al., 2015*). It has been shown that soluble forms of CD163 and CD206 can be shed from macrophages in contact with bacteria and fungi and promote opsonophagocytosis of *S. aureus* (*Kneidl et al., 2012*), *C. albicans* and *A. fumigatus* (*Gazi et al., 2011*). These observations imply that soluble forms of scavenger receptors may be generated under certain pathophysiological conditions. In agreement with this, we have previously demonstrated generation of soluble CL-12 from the cell-bound form *Ma et al., 2015*. The soluble CL-12 was found to retain the capability to recognize certain fungal pathogens such as *A. fumigatus, Lichtheimia corymbifera, Mucor circinelloides* etc (*Zhang et al., 2019*). Moreover, soluble CL-12 stimulated AP complement activation, most likely via properdin, but the mechanism of this observation was not investigated further.

Whether properdin can function as a PRM and initiate in situ assembly of C3bBb amplifying complement activation is controversial (*Harboe et al., 2017*). The function of properdin as a PRM has been mostly investigated either under the presence of possible contribution from serum-generated C3 activation or under conditions where non-physiological forms of properdin appears to initiate AP activation, Thus, it is still controversial to define properdin as a complement initiating molecule in addition to its C3bBb stabilizing function (*Cortes et al., 2011*; *Kemper et al., 2008*; *Xu et al., 2008*; *Saggu et al., 2013*; *O'Flynn et al., 2014*; *Wang et al., 2015*). Despite this contention, surface plasmon resonance analysis has clearly shown that properdin-immobilized biosensor chips can

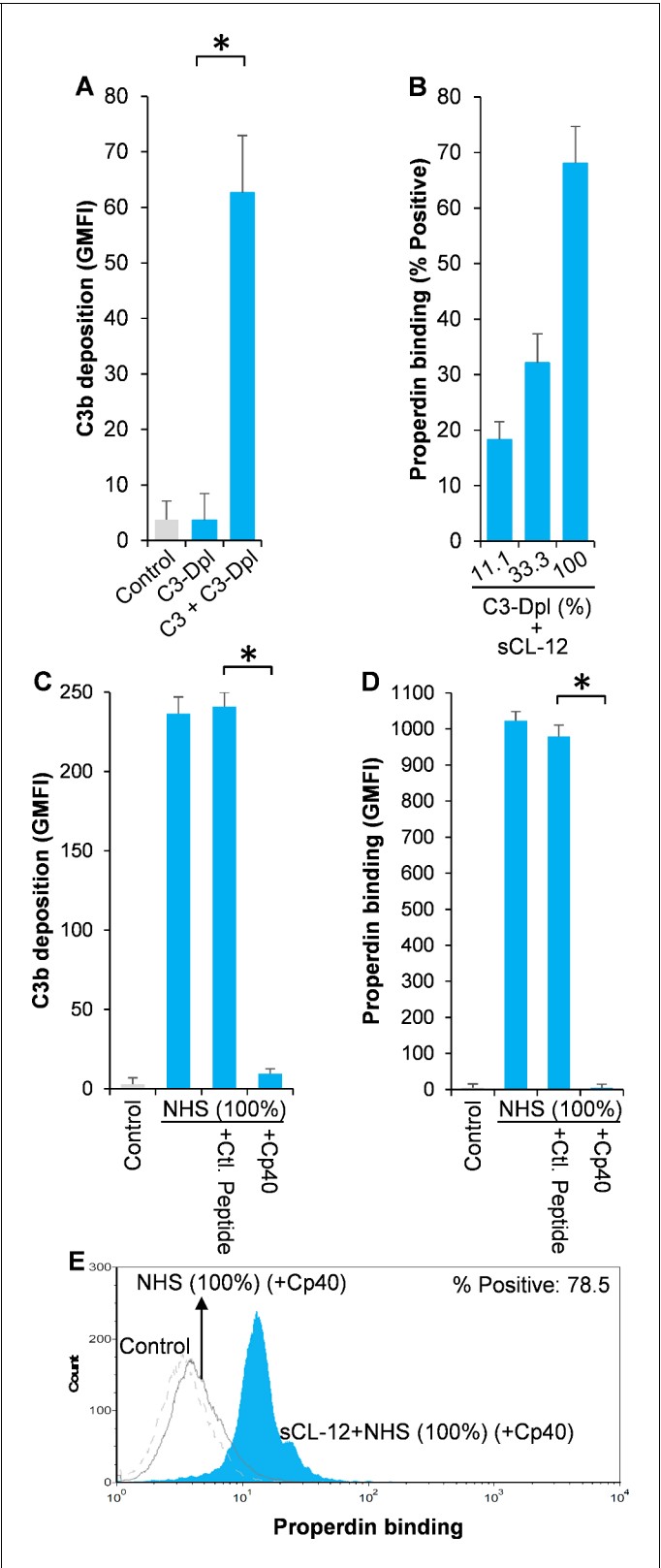

**Figure 4.** Effect of soluble CL-12 on properdin binding in C3-absent serum. *A. fumigatus* was incubated with C3-depleted serum (C3-Dpl) (10%) reconstituted with exogeneous C3 (130 µg/ml) (**A**). In some experiments, *A. fumigatus* was incubated with or without sCL-12 (5 µg/ml) prior to addition of C3-Dpl (11.1 ~ 100%) (**B**). Inhibitory effect of compstatin analog Cp40 (20 µM) was determined on NHS (100%)-induced complement activation as *Figure 4 continued on next page*

*Figure 4 continued*

described above. C3b deposition (**A, C**) or properdin binding (**B, D, E**) was analyzed and expressed as the GMFI or % positive events with respect to the C3-Dpl alone by flow cytometry. Results are representative of at least six independent experiments. *p<0.01.

The online version of this article includes the following figure supplement(s) for figure 4:

**Figure supplement 1.** Binding of properdin to *A.niger* in the presence of soluble CL-12 and compstatin analog Cp40.

**Figure supplement 2.** Soluble CL-12-dependent properdin binding on *A.niger*.

**Figure supplement 3.** Effect of soluble CL-12 on properdin binding in C3-absent serum.

initiate C3bBb assembly upon addition of purified C3b, factor B and factor D (*Hourcade, 2006*). However, extrapolating this finding to a biologically relevant condition has been disputed since properdin covalently linked to biosensor chip might be in a non-physiological aggregated form. Nevertheless, this finding might imply that given a proper 'sensory' input it seems plausible that properdin mediates de novo assembly of C3bBb. However, recent studies challenged and revised the view of properdin as a PRM, suggesting that properdin binding to the AP activator surfaces happens solely in a C3-dependent manner (*Harboe et al., 2012*; *Harboe et al., 2017*; *Agarwal et al., 2010*). In agreement with these observations, our data consistently show that properdin does not bind directly to Aspergillus strains, suggesting that properdin itself does not have potential to act as the AP initiating molecule on this AP activator matrix. However, our previous findings show that properdin can synergize the AP activation, C3 deposition and formation of terminal membrane attack complex on *A. fumigatus* in the presence of soluble CL-12 whereas no clear effect of CL-12 was found on C4 activation (*Ma et al., 2015*). By contrast, CL-12 could not recognize another opportunistic fungal pathogen *C. albicans*. Without CL-12 opsonization, complement activation is dependent on the lectin pathway via MBL and collectin-11 on *C. albicans* (*Ma et al., 2013a*). These findings imply the specific binding and exclusive effector mechanism of CL-12 against certain pathogens.

In the present study, we asked the question of whether properdin might promote AP activation specifically in a soluble CL-12-dependent manner independent of prior C3 deposition. Through control of initial C3 activation by the C3 blocker compstatin analog Cp40 and application of *Aspergillus* stains as a model of infection, we provide evidence that properdin binds to the fungal pathogens just after initial opsonization by soluble CL-12, but independent of C3, thus directing in situ assembly of C3bBb. Therefore, the evidence herein supports the conception that properdin itself may spark AP activation as a member of a hetero-PRMs complex comprising CL-12 as the PRM molecule binding in a C3-independent manner. In addition to complement amplification, the potential physiological consequences of the soluble CL-12:properdin crosstalk in host defense against infections of Aspergillus strains are intriguing and remain question. Previously, properdin binding to endothelial cells has been suggested, largely based on surface expression of heparin sulfate proteoglycans to attract properdin on vascular endothelial cells and renal tubular epithelial cells (*Zaferani et al., 2011*; *Mertens et al., 1992*). However, even though endothelial cells contribute as source of serum properdin and express heparin sulfate proteoglycans, no data exist to support direct binding of properdin to endothelial cells. Furthermore, a recent report has shown that the presence of the Cp40 can completely abolish properdin binding on human umbilical vein endothelial cells (HUVECs), suggesting that properdin binding is secondary to initial C3b deposition (*Harboe et al., 2017*). In this case, it is noteworthy that as a significant source of transmembrane scavenger receptor CL-12, vascular endothelial cells, for instance, HUVECs, are likely to express a considerable level of CL-12. In analogy with this finding, we did not observe any apparent binding of properdin to either CHO/CL-12FL, HUVECs or HUAECs, even though these cells were determined to express transmembrane CL-12 (*Ma et al., 2015*). However, our previous data clearly show that properdin could directly interact with soluble form of CL-12 in a microtiter plate and cell-based setup as well (*Ma et al., 2015*). Combining these data, it implies that CL-12 in its soluble form, when bound to a ligand, can attract properdin, but not when present as a membrane-anchored protein. Given the fact that there is considerable heterogeneity in properdin binding on soluble CL-12-opsonized *A. fumigatus* in the presence of the Cp40 and NHS, whether the soluble CL-12:properdin crosstalk additionally involves a complex mixture of different humoral immune components remains question.

Unlike most of the complement components which are predominantly expressed by hepatocytes, properdin is also produced by different leukocytes (*Schwaeble et al., 1994*; *Schwaeble et al.,*

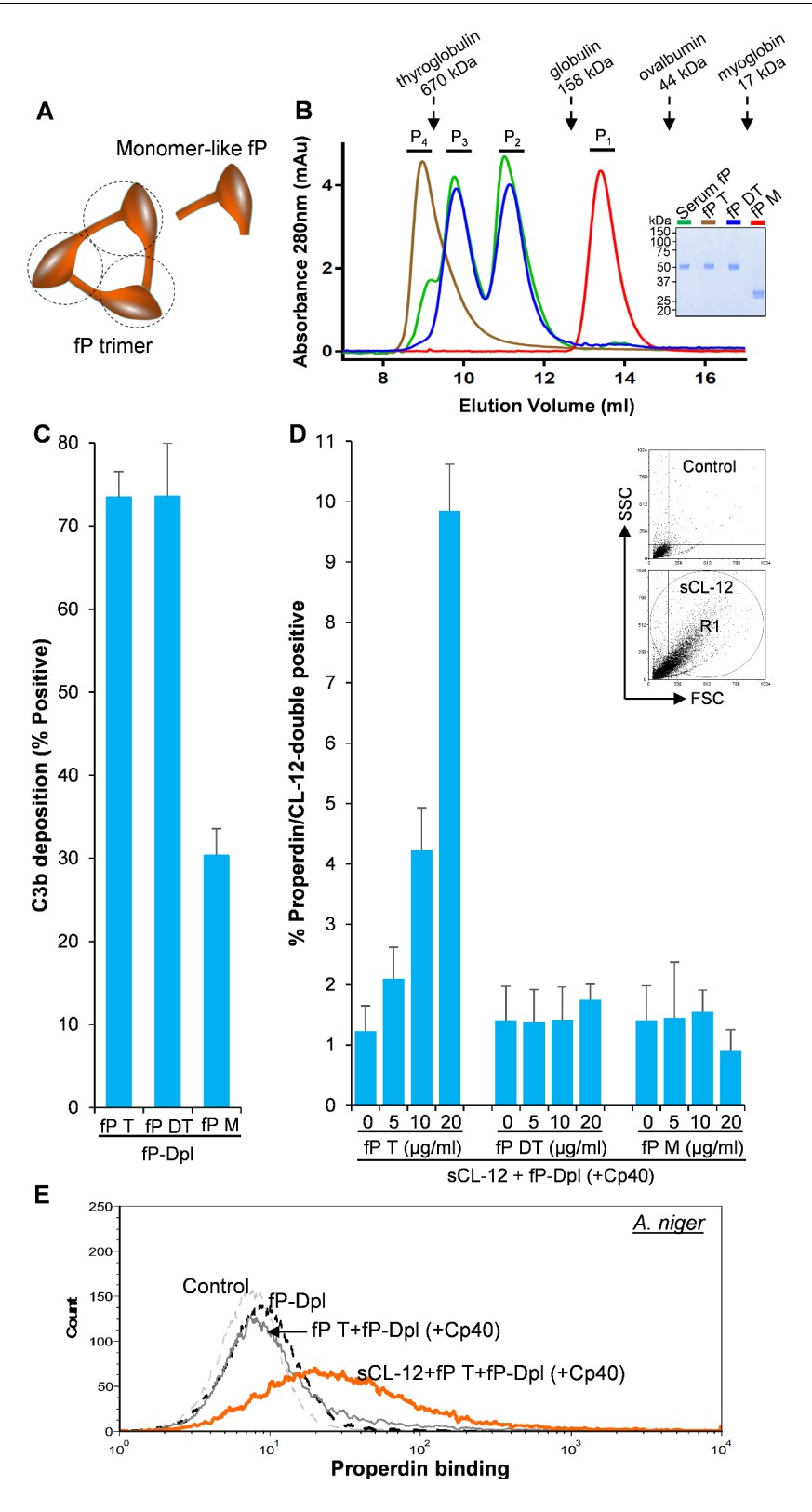

**Figure 5.** Binding specificity of soluble CL-12 towards properdin variants. (**A**) Illustrative diagram of monomer-like properdin complex. (**B**) SEC analysis of properdin. *Inset*: SDS-PAGE analysis of properdin under reducing condition. (**C**) *A. fumigatus* were incubated with fP-Dpl (10%) restored with purified serum properdin (10 μg/ml) or properdin variants (fP T: properdin tetramer; fP DT: properdin dimer/trimer; fP M: monomer-like properdin, 10 μg/ml). C3b deposition were analyzed and expressed as % Positive events with respect to the fP-Dpl alone with mean ± S.E.M from three independent

*Figure 5 continued on next page*

*Figure 5 continued*

experiments. (D) sCL-12 (5 µg/ml) was preincubated with the fungus prior to addition of the serum treated with Cp40 (6 µM) plus the properdin variants. Binding of properdin variants was analyzed in sCL-12-positive region (R1 in FSC vs SSC of the fungus) and expressed as % properdin/CL-12-double positive events with mean ± S.E.M from three independent experiments. (E) Binding of properdin tetramer (20 µg/ml) was determined on *A. niger* as above. Results are representative of at least six independent experiments.

The online version of this article includes the following figure supplement(s) for figure 5:

**Figure supplement 1.** Binding of properdin variants to *A.niger* in the presence of soluble CL-12.

---

*1993*; *Wirthmueller et al., 1997*; *Stover et al., 2008*). However, leukocytes-derived properdin is ~100 times more active than serum-derived properdin in an AP hemolytic assay (*Schwaeble et al., 1993*). Whether the molecular composition and function of leukocyte produced properdin resemble serum properdin remains unknown. Recently, Harboe et al. observed that properdin reconstituted to properdin-deficient serum bound HUVECs in an initial C3b-dependent manner, but not detected with properdin in C3-deficient serum (*Harboe et al., 2017*). Consistent with this observation, we found that binding of isolated properdin was remarkably reduced in the presence of properdin-deficient serum on *A. fumigatus* and further abolished in the presence of the Cp40. Similar results were also achieved when properdin variants (fP T, fP DT or fP M) were applied. These results suggest that unknown serum factors may compete or inhibit properdin binding to the AP activator surfaces. It might be hypothesized that this may protect self-tissue from constitutive complement-mediated damage, as has been suggested for serum amyloid P component (SAP), which limits properdin function in the presence of serum (*Mitchell and Hourcade, 2008*). Given the fact that various leukocytes express properdin, locally released properdin could bypass regulation by serum factors, thus exert differential functions at the local tissue level.

Multiple studies have suggested the essential role of properdin oligomerization for complement activation and amplification, and the tetramer has repeatedly been found to be significantly more active than both the dimer and the trimer among the physiological forms of properdin (*Pedersen et al., 2017*; *Pangburn, 1989*; *Blatt et al., 2016*; *Ali et al., 2014*). We have previously shown that when properdin was prevented from forming oligomers and only formed the monomeric form, it was unable to support complement-mediated bacteriolysis and with a strong reduction in the capability of supporting C3b amplification (*Pedersen et al., 2017*). These findings further support the importance of properdin oligomerization for its in vivo function. Given the fact that properdin exists as approx. 30% dimers, 50% trimers and 20% tetramers in serum (*Harboe et al., 2017*; *Pedersen et al., 2017*; *Pangburn, 1989*), our data imply that soluble CL-12 is likely to preferably recognize properdin organized as higher oligomer which may increase its avidity to interact with soluble CL-12 or/and fluid-phase C3 convertase simultaneously. Our results with the fP T, the fP DT and the fP M further substantiate this is indeed the case. Properdin is stored in mast cells and neutrophils (*Wirthmueller et al., 1997*; *Stover et al., 2008*), which location are closely associated with blood vessels and thus rather convenient to play a sentinel role in innate immune defense. Therefore, such locally released properdin might have differences in oligomerization and posttranslational modifications compared to its serum form, thus representing a more active form with a higher affinity towards binding partners.

In conclusion, we provide evidence that it is necessary to revise the notion about properdin showing that it can function in complex with a PRM as shown here with the example of CL-12. We argue that properdin also may specifically direct AP of complement activation via sensory inputs from CL-12 independently of primary C3 deposition in addition to its well-described stabilizing effect of the AP C3 convertase.

## Materials and methods

### Cell lines

HEK293F (Source: Gibco, Identifier: R79007) was involved in our work. Each lot of Gibco 293 F cells was tested for cell growth and viability post-recovery from cryopreservation. The Master Seed Bank has been tested for contamination of bacteria, yeast, mycoplasma and virus and has been characterized by isozyme and karyotype analysis.

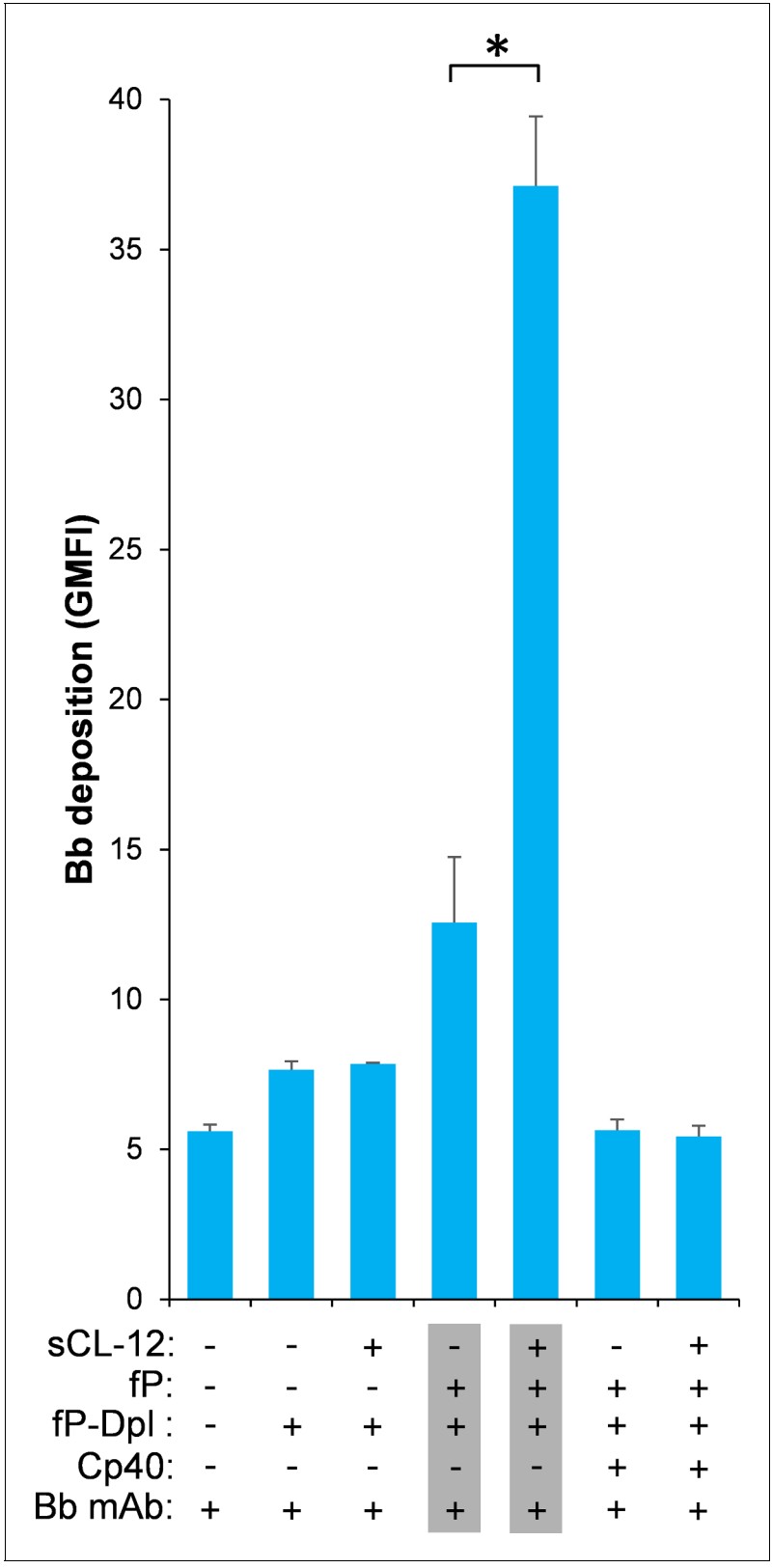

**Figure 6.** Effect of soluble CL-12 on properdin-directed AP C3 convertase assembly on *A.fumigatus. A. fumigatus* were incubated with fP-Dpl (10%) in the presence or absence of sCL-12 (5 µg/ml) as described above. In some experiments, the serum was restored with serum properdin (10 µg/ml) in the presence of Cp40 (6 µM). Bb deposition was analyzed by flow cytometry. The GMFI was used to assess protein binding and expressed as mean ± S.E.M from three independent experiments. Results are representative of at least six independent experiments. *p<0.01.

*Figure 6 continued on next page*

*Figure 6 continued*

The online version of this article includes the following figure supplement(s) for figure 6:

**Figure supplement 1.** Effect of soluble CL-12 on properdin-directed AP C3 convertase assembly on *A.niger*.

## Production of recombinant properdin

DNA encoding human properdin was synthesized (GenScript) with the endogenous signaling peptide and a C-terminal 6xHis-tag and cloned into the pCEP4 mammalian expression vector using HindIII and BamHI restriction sites. Oligomeric properdin (fP T or fP DT) was expressed by transient

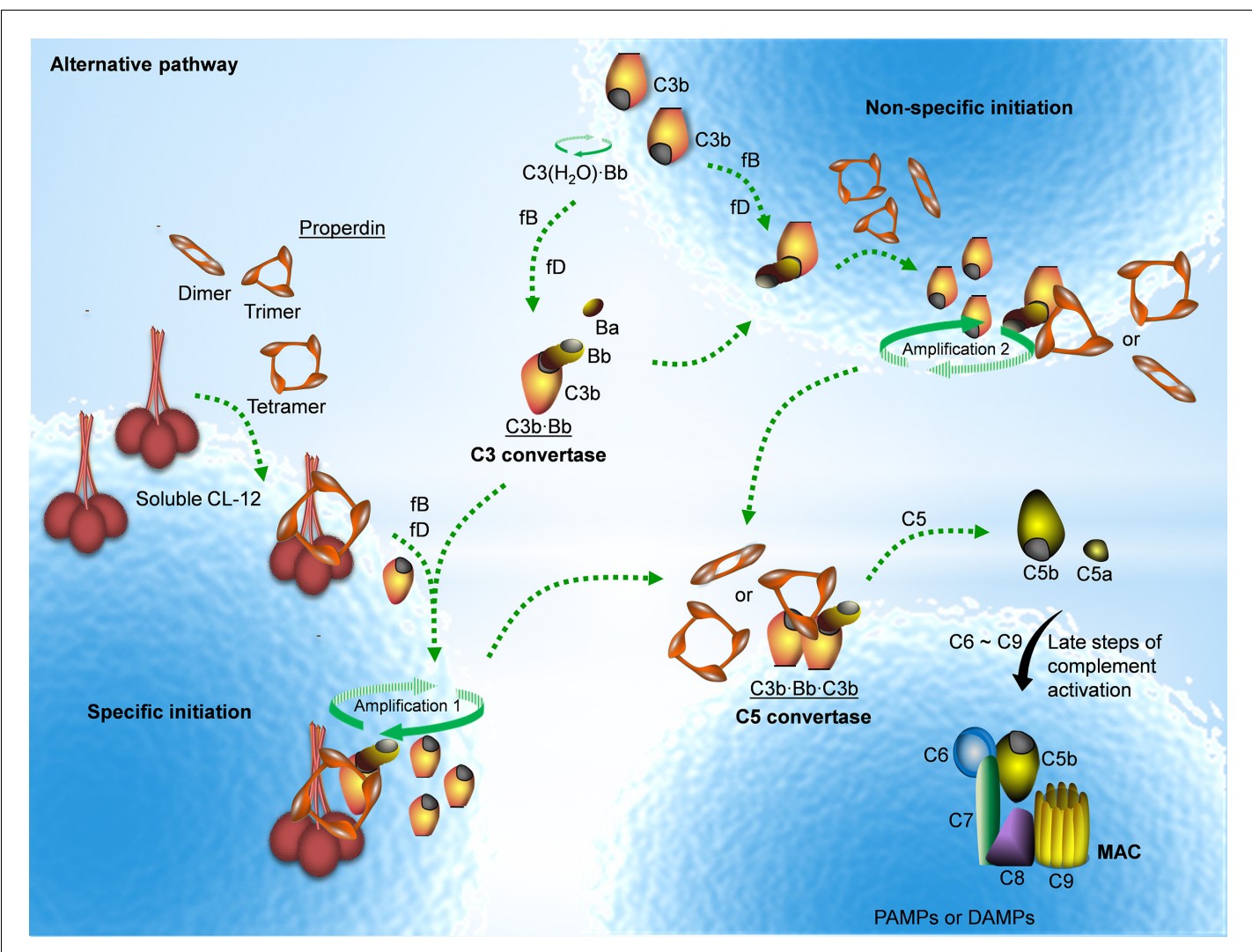

**Figure 7.** Proposed avenues of alternative pathway activation. Specific initiation: In addition to the non-specific AP activation, the AP is also specifically triggered by pattern- recognition molecules (PRMs), as soluble CL-12. Upon opsonization of target surfaces, soluble CL-12 specifically recruits properdin organized as higher oligomer (ex. tetramer) to initiate in situ assembly of C3bBb or promotes surface recruitment of preformed fluid-phase C3bBb. When the constant turnover of the in situ assembly drives around, C3b production is highly amplified (amplification 1). Non-specific initiation: AP activation often occurs when C3 thioester is spontaneously activated in the fluid phase and forms initiation C3 convertase C3(H₂O)Bb on nearby AP activator surfaces. The initiation is independent of the trigger, and starts to create AP C3 convertase (C3bBb) based on the low level of non-specifically anchored nascent C3b in collaboration with factor B (fB) and factor D (fD). When the AP C3 convertase is stabilized through the association of the complement positive regulator properdin, the productivity of C3b is highly amplified (amplification 2). Both avenues of the specific and non-specific responses contribute to the feedback C3 amplification loop, which substantially potentiates formation of AP C3 convertase, C5 convertase (C3bBbC3b) and C5b-9 membrane attack complex (MAC) for complement activation.

transfection (*Pedersen et al., 2019b*) and monomeric properdin (monomer-like properdin complex) was generated by co-transfection of properdin head fragment (TB-TSR3) and tail fragment (TSR4-TSR6) as described previously (*Pedersen et al., 2019a*). Oligomeric and monomeric properdin were expressed in HEK293F cells as described previously (*Pedersen et al., 2019b*). fP T and fP DT were purified by gel filtration using a 24 ml Superdex 200 increase column (GE Healthcare) equilibrated in equilibration buffer (20 mM HEPES/150 mM NaCl, pH 7.5).

## Compstatin analog Cp40 and its control peptide

Compstatin analog Cp40 (dTyr-Ile-[Cys-Val-Trp(Me)-Gln-Asp-Trp-Sar-His-Arg-Cys]-mIle-NH$_2$, 1.7 kDa) was produced by our group at the University of Pennsylvania and applied to specifically block C3 activation (*Risitano et al., 2014*). Negative control peptide to the Cp40 (Ile-Ala-Val-Val-Gln-Asp-Trp-Gly-His-His-Arg-Ala-Thr-NH$_2$, 1.5 kDa) was obtained from Tocris Bioscience.

## Efficacy of compstatin analog Cp40 on C3b deposition

Normal human serum (NHS) (20%) (from healthy volunteer blood donors with informed consent) was preincubated with the compstatin analog Cp40 (0–6 µM) or the control peptide (0–6 µM) for 10 min prior to incubation with *Aspergillus fumigatus* (*A. fumigatus*). Complement C3b deposition was then determined in the presence of TBS/Ca$^{2+}$ as previously described (*Ma et al., 2015*), and analysed by flow cytometry. Data were evaluated using FCS express version 5 (De Novo software).

## Preparation of fungal pathogens

Isolates of *A. fumigatus* strain was obtained from fatal clinical specimens (No. 6871) of invasive pulmonary aspergillosis (IPA) (*Naaraayan et al., 2015*) (a kind gift from Professor Luigina Romani, Microbiology Section, Department of Experimental Medicine and Biochemical Science, University of Perugia). Isolates of *A. niger* strain (N32) was obtained from Professor Reinhard Würzner (Innsbruck Medical University, Innsbruck). Resting conidia were prepared as previously described (*Ma et al., 2015*; *Zhang et al., 2019*). In brief, the *Aspergillus* strains were grown on sabouraud dextrose agar supplemented with chloramphenicol by agar-streak for 4 days at 28˚C. Abundant conidia were obtained under these conditions. The conidia were harvested by washing the slant culture with PBS supplemented with 0.025% Tween 20 (PBS-T) and gently scraping the conidia from the mycelium with sterile cotton-tipped applicators. The conidia were then allowed to settle by gravity, followed by filtration through sterile 40 µM cell strainer to remove hyphal fragments and cell debris. After extensive washing with PBS-T, the conidia were counted and diluted to the desired concentrations. The conidia were heat-inactivated at 121˚C for 15 min prior to use. Clinically isolated *C. albicans* H29929 was obtained and prepared as described previously (*Ma et al., 2011*).

## Binding of soluble CL-12 to *A. fumigatus*, *A. niger* and *C. albicans*

Binding of soluble CL-12 to *A. fumigatus* was performed as previously described (*Ma et al., 2009*; *Ma et al., 2013b*). Aspergillus strains were incubated with recombinant soluble CL-12 (5 µg/ml, R and D system). Bound proteins were detected with CL-12 pAb one and Alexa Fluor 488-conjugated donkey anti-goat IgG (Invitrogen), and finally analyzed by flow cytometry. Data were evaluated using FCS express version 5 (De Novo software). In some experiments, the binding was also determined on *A. niger* or *C. albicans* as described previously (*Zhang et al., 2019*; *Ma et al., 2011*).

## Binding of properdin to *A. fumigatus* and *A. niger*

*A. fumigatus* were incubated with NHS (20%) treated with or without compstatin analog Cp40 (6 µM) or control peptide (6 µM) under TBS/EGTA-Mg$^{2+}$ (10 mM Tris/150 mM NaCl/10 mM EGTA/5 mM Mg$^{2+}$, pH7.4) as described above. When incubation with 100% NHS instead, the Cp40 or control peptide was treated with 20 µM. After washing, properdin binding was detected by sheep anti-human properdin pAb (R and D system) and Alexa fluor 488-conjugated donkey anti-sheep IgG (Invitrogen). Bound proteins were analyzed by flow cytometry as described above. In some experiments, *A. fumigatus* were preincubated with soluble CL-12 (5 µg/ml) prior to addition of NHS (20% or 100%), properdin-depleted human serum (fP-Dpl) (10%, CompTech) with or without addition of serum properdin (10 µg/ml, CompTech) or C3-depleted human serum (C3-Dpl) (0 ~ 100%, CompTech), and assessed as described above. The binding specificity of soluble CL-12 towards properdin

oligomers was tested with fP T, fP DT or fP M instead. Alternatively, the binding was also determined on *A. niger* as described above.

Serum properdin (CompTech) was thawed once from initial preparation, and preparations that formed aggregates during storage were excluded from testing through SEC analysis prior to use. Serum properdin was organized mainly as physiological oligomers (dimers, trimers and tetramers) under the conditions above (*Harboe et al., 2017*; *Pedersen et al., 2017*). Purity and oligomeric status of fP T, fP DT and fP M were determined by SDS-PAGE and SEC analysis prior to use as described above. In some experiments, C3-Dpl (10%) was spiked with exogenous C3 (130 µg/ml, CompTech) under physiological concentration, and assessed complement activity by measurement of C3b deposition as described above. The level of properdin was quantified to be ~14.6 µg/ml in the C3-Dpl (*Harboe et al., 2017*).

### Deposition of Bb on *A. fumigatus* and *A. niger*

*A. fumigatus* was incubated with properdin-restored fP-Dpl in the presence or absence of soluble CL-12 (5 µg/ml) as described above. Bb deposition was detected by mouse anti-human Bb mAb (Quidel) and FITC-conjugated goat anti-mouse IgG (Dako) and analyzed by flow cytometry as described above. Alternatively, the binding was also determined on *A. niger* as described above.

### Statistical analysis

Data were analyzed by GraphPad Prism version 8.0 (GraphPad Software) for statistics. Significant differences were analyzed by one-way ANOVA to compare between control and treated samples. *P* values < 0.01 were considered statistically significant.

## Acknowledgements

This work was supported by grants from the Kirsten og Freddy Johansens Fond, the Weimann and Seedorffs Foundation, the Benzon Foundation, The Novo Nordisk Research Foundation and the Danish Research Foundation of Independent Research [DFF-6110–00489].

## Additional information

### Competing interests

John D Lambris: inventor of patents (Patent Number: 9630992) and/or patent applications 426 (Application Number: 15/126,937) that describe the use of complement inhibitors for therapeutic purposes, founder of Amyndas Pharmaceuticals, which is developing complement inhibitors (i.e., third-generation compstatins) for clinical applications, and inventor of the compstatin technology licensed to Apellis Pharmaceuticals (i.e., 4[1MeW]7W/POT-4/APL-1 and PEGylated derivatives). The other authors declare that no competing interests exist.

### Funding

| Funder | Grant reference number | Author |
| --- | --- | --- |
| Kirsten og Freddy Johansens Fond | Research fund | Ying Jie Ma |
| Novo Nordisk | Research fund | Peter Garred |
| Købmand I Odense Johan og Hanne Weimann Født Seedorffs Legat | Research fund | Ying Jie Ma |
| Alfred Benzon Foundation | Research fund | Ying Jie Ma Peter Garred |

The funders had no role in study design, data collection and interpretation, or the decision to submit the work for publication.

## Author contributions
Jie Zhang, Lihong Song, Anna Li, Investigation; Dennis V Pedersen, John D Lambris, Gregers Rom Andersen, Resources, Validation; Tom Eirik Mollnes, Resources, Software, Validation; Ying Jie Ma, Conceptualization, Supervision, Funding acquisition, Writing - original draft, Project administration, Writing - review and editing; Peter Garred, Conceptualization, Formal analysis, Funding acquisition, Writing - original draft, Writing - review and editing

## Author ORCIDs
Jie Zhang (iD) http://orcid.org/0000-0002-4472-3468
Gregers Rom Andersen (iD) http://orcid.org/0000-0001-6292-3319
Ying Jie Ma (iD) https://orcid.org/0000-0003-4003-2579

## Decision letter and Author response
Decision letter https://doi.org/10.7554/eLife.60908.sa1
Author response https://doi.org/10.7554/eLife.60908.sa2

# Additional files

## Supplementary files
• Transparent reporting form

## Data availability
All data generated or analysed during this study are included in the manuscript and supporting files.

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
