## [Decision Letter]

**Acceptance summary:**

This manuscript provides novel evidence that docking of properdin to Aspergillus can be independent of C3 when soluble collectin-12 (CL^-^12) is present on its surface. It shows in a systematic set of experiments that the docking of properdin can be independent of C3 by using serum, depleted serum, C3 inhibitor and reconstitution experiments. It shows that properdin in concert with CL^-^12 can direct AP activation.

**Decision letter after peer review:**

Thank you for submitting your article "Soluble collectin-12 mediates C3-independent docking of properdin that activates the alternative pathway of complement" for consideration by *eLife*. Your article has been reviewed by two peer reviewers, including Frank L van de Veerdonk as the Reviewing Editor and Reviewer #1, and the evaluation has been overseen by Carla Rothlin as the Senior Editor.

The reviewers have discussed the reviews with one another and the Reviewing Editor has drafted this decision to help you prepare a revised submission.

The authors provide evidence that docking of properdin to heat inactivated Aspergillus can be independent of C3 when collectin-12 is present on its surface. It is indeed a controversial topic whether properdin can act independently of C3 to direct AP activation. The authors provide a systematic set of experiments that provide evidence that this can be C3 independent using serum, depleted serum, C3 inhibitor and reconstitution experiments. The set of experiments are straightforward and sufficient to conclude that there are conditions where properdin in concert with CL^-^12 can direct AP activation.

Suggestions for revision:

1) The authors use aspergillus as a vehicle to study properdin and CL^-^12 interactions. Although it might be relevant for aspergillus infection eventually no data are going into detail of the role of properdin and Cl^-^12 in aspergillus or what kind of components are involved. Therefore the authors should state more clearly in the paper that Aspergillus is used as a model to study their hypothesis and has so not provided any insight or details on its role during Aspergillus infection.

2) The authors should remove the paragraph in the Introduction of the paper on Aspergillus infection and remove reference throughout the manuscript on the role complement system in aspergillus host defense. It is fine to discuss this in the Discussion, however it would be more clear for the reader when it is explained that aspergillus is just a model to explore their hypothesis. Then also the observation of the fact that Candida does not do it only provides evidence that it is not ubiquity doing this on all structures that express complex carbohydrates, but there are no answers on the specific ligands on the fungi that are responsible for this.

3) Direct evidence showing sCL^-^12 and properdin interaction is lacking, as normal human serum (NHS), sera depleted of specific component or reconstituted sera have been used throughout this study. As serum is a complex mixture of different humoral immune components, there might be multiple mechanisms responsible for the observed interaction between soluble CL^-^12 and properdin. Might even still be an indirect effect, but at least independent of

This should be discussed in more detail in the Discussion.

4) What is the sCL^-^12 concentration in normal human serum (NHS)?

5) Through the study either NHS or specific-component depleted sera are used; they are a complex mixture of different components. The authors need to validate their observation with individual/purified components. Specifically, in Figure 2C, in the spectra for properdin in the presence of Cp40 shows multiple peaks, which may represent multiple mechanisms of sCL^-^12 and properdin binding, as well other serum factors facilitating their binding.

6) Hourcade, 2006, shows that the properdin-immobilized biosensor chips can initiate C3bBb assembly when added with purified C3b, factor B and factor D. Nevertheless, what is its biological significance, as soluble C3b formed will be degraded further by Factors H and I?

---

## [Author Response]

Suggestions for revision:1) The authors use aspergillus as a vehicle to study properdin and CL^-^12 interactions. Although it might be relevant for aspergillus infection eventually no data are going into detail of the role of properdin and Cl^-^12 in aspergillus or what kind of components are involved. Therefore the authors should state more clearly in the paper that Aspergillus is used as a model to study their hypothesis and has so not provided any insight or details on its role during Aspergillus infection.

According to the reviewer’s instruction, we have revised the manuscript accordingly (subsections “C3b and properdin deposition on *A. fumigatus* incubated in normal human serum was C3-dependent” and “The binding of properdin to CL-12-opsonized fungi was specifically C3-independent” and the Discussion).

2) The authors should remove the paragraph in the Introduction of the paper on Aspergillus infection and remove reference throughout the manuscript on the role complement system in aspergillus host defense. It is fine to discuss this in the Discussion, however it would be more clear for the reader when it is explained that aspergillus is just a model to explore their hypothesis. Then also the observation of the fact that Candida does not do it only provides evidence that it is not ubiquity doing this on all structures that express complex carbohydrates, but there are no answers on the specific ligands on the fungi that are responsible for this.

We have followed the reviewer’s instruction and revised the manuscript accordingly (Introduction). Interestingly, our recent data revealed an unique pattern molecule on the fungi that might be responsible for specific sCL^-^12 opsonization and thus deploy certain immune responses. This is a subject of our current investigation.

3) Direct evidence showing sCL^-^12 and properdin interaction is lacking, as normal human serum (NHS), sera depleted of specific component or reconstituted sera have been used throughout this study. As serum is a complex mixture of different humoral immune components, there might be multiple mechanisms responsible for the observed interaction between soluble CL^-^12 and properdin. This should be discussed in more detail in the Discussion.

We have verified direct interaction between sCL^-^12 and properdin in cell-based context and microtiter plate. In both FACS and ELISA analysis, properdin bound immobilized sCL^-^12 in a dose-dependent manner and vice versa. Furthermore, we have also confirmed the specificity using recombinant CL^-^12 constructs (CL^-^12 full length and CL^-^12 extracellular domain) made in-house, indicating that properdin exclusively binds CL^-^12 when it becomes opsonin-like soluble lectin [Ma et al., 2015]. This was also validated in our recent data with vascular endothelial cells (ex. HUVECs or HUAECs) expressing transmembrane CL^-^12, suggesting that properdin could only interact with sCL^-^12 [Harboe et al., 2017; Ma et al., 2015].

Our data with the properdin variants also suggest that sCL^-^12 is not able to recognize certain TSR domains of single properdin, but preferably recognize properdin with tetramer organization, leading to in situ C3bBb assembly. Nevertheless, in agreement with the reviewer’s opinion, our data do not rule out the possibility of indirect sCL^-^12:properdin crosstalk via uncertain serum factors as shown in the case of MBL:PTX3:C1q heterocomplexes as well as MBL:SAP: X serum factor(s) heterocomplexes that amplify complement activation and synergize immune effector functions [Ma et al., 2011]. (We have followed the reviewer’s instruction and revised the manuscript accordingly. See the Discussion).

4) What is the sCL^-^12 concentration in normal human serum (NHS)?

sCL^-^12 was previously shown to circulate in normal human umbilical cord plasma at median conc. of <inline-graphic mime-subtype="png" mimetype="image" xlink:href="media/image1.png" />92.8ng/ml, but not clearly detected in normal adult venous plasma or serum [Ma et al., 2015]. Several lines of recent evidence suggest that CL^-^12 are not only expressed on the human umbilical cord endothelium, but also on the surface of phagocytes (ex. alveolar macrophages, central nervous system resident macrophages and microglia cells) [Selman et al., 2008; Bogie et al., 2017; Chang et al., 2018].

These studies suggest the essential involvement of CL^-^12 in immune recognition of DAMPs and likely generation of sCL^-^12 in response to inflammatory signals during various processes of immune responses, as has been shown for soluble CD163 and a more recent finding of soluble CD206 [Etzerodt et al., 2013; Rodgaard-Hansen et al., 2014]. In support of the previous findings, we also found soluble CL^-^12 expression in CSF of healthy individuals as well as in the plasma of sepsis patients. Therefore, sCL^-^12 can be generated and the local level might be highly elevated under certain pathophysiological conditions.

5) Through the study either NHS or specific-component depleted sera are used; they are a complex mixture of different components. The authors need to validate their observation with individual/purified components. Specifically, in Figure 2C, in the spectra for properdin in the presence of Cp40 shows multiple peaks, which may represent multiple mechanisms of sCL^-^12 and properdin binding, as well other serum factors facilitating their binding.

We have validated our observations by using purified components, and further confirmed the specificity using recombinant CL^-^12 constructs (CL^-^12 full length and CL^-^12 extracellular domain) made in-house, suggesting that properdin exclusively binds CL^-^12 when it becomes opsonin-like soluble lectin [Ma et al., 2015]. Our data with the properdin variants also suggest that properdin binds sCL^-^12 in a C3-independent manner exclusively via its tetrameric structure. Nevertheless, in agreement with the reviewer’s opinion, our data do not rule out the possibility of indirect sCL^-^12:properdin crosstalk via uncertain serum factors as shown in the case of MBL:PTX3:C1q heterocomplexes as well as MBL:SAP: X serum factor(s) heterocomplexes that amplify complement activation and synergize immune effector functions [Ma et al., 2011].

6) Hourcade, 2006, shows that the properdin-immobilized biosensor chips can initiate C3bBb assembly when added with purified C3b, factor B and factor D. Nevertheless, what is its biological significance, as soluble C3b formed will be degraded further by Factors H and I?

We totally agree with reviewer’s opinion. Recent studies utilizing surface plasmon resonance methodology observed that properdin covalently attached to a biosensor can direct the assembly of the properdin·C3b·Bb in a step-wise fashion (Hourcade, 2006). Although extrapolating this finding to a biologically relevant condition has been disputed due to the use of the artefactual platform, degradation of soluble C3b by fH/fI as well as properdin covalently linked to biosensor chip might be in a non-physiological aggregated form. Nevertheless, this finding might imply that given a proper “sensory” input it seems plausible that properdin mediates de novo assembly of C3bBb.